# Elevated blood remnant cholesterol and triglycerides are causally related to the risks of cardiometabolic multimorbidity

Yimin Zhao [1,2], Zhenhuang Zhuang[2], Yueying Li [2], Wendi Xiao[2], Zimin Song[2], Ninghao Huang[2], Wenxiu Wang[2], Xue Dong[2], Jinzhu Jia[3], Robert Clarke [4] & Tao Huang [2,5,6] ✉

The connection between triglyceride-rich lipoproteins and cardiometabolic multimorbidity, characterized by the concurrence of at least two of type 2 diabetes, ischemic heart disease, and stroke, has not been definitively established. We aim to examine the prospective associations between serum remnant cholesterol, triglycerides, and the risks of progression from first cardiometabolic disease to multimorbidity via multistate modeling in the UK Biobank. We also evaluate the causality of these associations via Mendelian randomization using 13 biologically relevant SNPs as the genetic instruments. Here we show that elevated remnant cholesterol and triglycerides are significantly associated with gradually higher risks of cardiometabolic multimorbidity, particularly the progression of ischemic heart disease to the multimorbidity of ischemic heart disease and type 2 diabetes. These results advocate for effective management of remnant cholesterol and triglycerides as a potential strategy in mitigating the risks of cardiometabolic multimorbidity.

Blood lipids include low-density lipoprotein cholesterol (LDL-C), high-density lipoprotein cholesterol (HDL-C), triglycerides, and lipoprotein(a). Despite consistent and robust evidence underscoring the deleterious effects of LDL-C on the risk of atherosclerotic cardiovascular disease, the independent relevance of triglyceride-rich lipoproteins remains uncertain. Previous studies elucidating the adverse effects of triglycerides on cardiovascular disease were hampered by the substantial inverse associations between blood levels of triglycerides and HDL-C[1]. Thus, hypertriglyceridemia has traditionally been considered as a biomarker or bystander for low blood HDL-C rather than an intrinsic risk factor[1]. Recent advances in human genetics and unanticipated outcomes from clinical trials investigating the efficacy of HDL-C-raising medication have reshaped the importance of triglycerides and HDL-C[2-4]. Elevated triglycerides were causally associated with a higher incidence of heart disease and aortic stenosis in Danish adults[5-7]. Mendelian randomization analyzes also provided genetic support indicating that elevated plasma triglycerides were superior therapeutic targets to HDL-C, despite the absence of any licensed effective triglyceride-lowering treatment beyond fibrates for heart disease prevention[8,9].

Cardiometabolic diseases, including type 2 diabetes, ischemic heart disease, and stroke, remain the primary causes of premature death worldwide[10]. Moreover, cardiometabolic multimorbidity, characterized by the concurrence of at least two cardiometabolic diseases, has an adverse effect on the quality of life and two-fold higher risks of premature death than either of these diseases alone[11]. Given their role

[1]Department of Sports Medicine, Peking University Third Hospital, Peking University, Beijing, China. [2]Department of Epidemiology and Biostatistics, School of Public Health, Peking University, Beijing, China. [3]Department of Biostatistics, School of Public Health, Peking University, Beijing, China. [4]Clinical Trial Service Unit and Epidemiological Studies Unit, Nuffield Department of Population Health, University of Oxford, Oxford, United Kingdom. [5]Key Laboratory of Epidemiology of Major Diseases (Peking University), Ministry of Education, Beijing, China. [6]Center for Intelligent Public Health, Academy for Artificial Intelligence, Peking University, Beijing, China. ✉e-mail: huangtaotao@pku.edu.cn

in the onset of different cardiometabolic diseases and their progression to multimorbidity, higher levels of triglyceride-rich lipoproteins have been suggested as a risk factor for multimorbidity[12–14]. Inefficient lipoprotein lipase (LPL)-mediated lipolysis and hepatic over-production of very low-density lipoprotein (VLDL) result in accumulation of triglyceride-rich lipoprotein remnants in blood. Importantly, the remnants of triglyceride-rich lipoprotein contain about four-fold higher cholesterol per particle than LDL-C and are small enough to penetrate the endothelial barrier, which was believed to be more atherogenic than LDL-C[15,16]. Unlike LDL particles, cholesterol in triglyceride-rich lipoprotein remnants could be directly taken up by macrophages, facilitating the formation of foam cells and, subsequently, atherosclerotic plaque. In addition, remnant cholesterol but not LDL-C, was causally linked to low-grade inflammation[17]. Recent studies have reported the causal relevance of triglyceride-rich lipoproteins and their remnants for the risk of cardiovascular disease[6,16,18]. However, the importance of triglyceride-rich lipoproteins, namely remnant cholesterol and triglycerides, for progression from first cardiometabolic disease to multimorbidity is not fully understood. It is noteworthy that diet, like fat and cholesterol intake, and lifestyle factors might substantially influence triglyceride-rich lipoprotein metabolism[19–21]. The complex interplay between lipid species might also confound the observational relationships between triglyceride-rich lipoproteins and cardiometabolic diseases[22].

In this study, we utilized multistate modeling and two-stage least squares regression-based Mendelian randomization to investigate the associations between remnant cholesterol, triglycerides and the risks of cardiometabolic multimorbidity in the UK Biobank. We found that blood remnant cholesterol and triglycerides were causally associated with higher risks of progression from first cardiometabolic disease to multimorbidity, particularly the multimorbidity of ischemic heart disease and type 2 diabetes.

## Results

### Baseline characteristics of study participants

Table 1 shows the baseline characteristics of participants according to serum concentrations of remnant cholesterol and triglycerides. Participants with higher serum levels of remnant cholesterol and triglycerides tended to be men and older, have higher body mass index and systolic blood pressure, have more vulnerable socioeconomic status, be current or former smokers, have a more extended smoking history, favor access alcohol intake, be physically inactive, and have less healthy diet and sleep quality (Table 1). In contrast, these covariates were more evenly distributed when stratified by the genetic risk score (GRS) quintiles (Supplementary Data 1). The average serum concentrations of remnant cholesterol and triglycerides in the entire study population were 0.70 mmol/L (median: 0.67 mmol/L; interquartile range: 0.50, 0.86 mmol/L) and 1.70 mmol/L (median: 1.44 mmol/L; interquartile range: 1.02, 2.08 mmol/L), respectively. Serum concentrations of remnant cholesterol and triglycerides were similar among participants with different fasting time (Supplementary Data 2).

### Observational analyzes

During a median follow-up of 12.5 years (4,694,404 total person-years), we identified 39,084 newly onset first cardiometabolic disease cases among 391,583 participants that were free of any cardiometabolic diseases at baseline (Supplementary Fig. 1a). Among people that experienced any cardiometabolic diseases, 3794 subsequently progressed to cardiometabolic multimorbidity. Specifically, 2370 participants further developed the co-morbidity of ischemic heart disease and type 2 diabetes, also termed IHD-T2D multimorbidity (Supplementary Fig. 1b).

We first pooled all three cardiometabolic disease together and found that elevated serum triglycerides were associated with higher risks of incident cardiometabolic disease and progression of first cardiometabolic disease to multimorbidity (Supplementary Fig. 1a). Compared with participants with serum triglycerides <0.9 mmol/L, the multivariate-adjusted hazard ratios (HRs) for progression from first cardiometabolic disease to cardiometabolic multimorbidity were 1.28 (95% CI: 1.09, 1.51), 1.29 (1.10, 1.52), 1.36 (1.16, 1.59), and 1.50 (1.29, 1.75) for those with serum triglycerides 0.9–<1.3 mmol/L, 1.3–<1.7 mmol/L, 1.7–<2.3 mmol/L, and ≥ 2.3 mmol/L, respectively (Table 2). However, when separating the cardiometabolic diseases into three distinct endpoints, it was observed that serum remnant cholesterol and triglycerides were exclusively associated with elevated risks of ischemic heart disease and type 2 diabetes, but not stroke (Supplementary Data 3).

We next tested whether serum remnant cholesterol and triglycerides involved in the trajectories from disease-free status to ischemic heart disease/type 2 diabetes and to subsequent IHD-T2D multimorbidity (Supplementary Fig. 1b). Participants with serum remnant cholesterol of ≥1.0 mmol/L had 146% (HR: 2.46; 95% CI: 2.23, 2.71; P for trend <0.001) and 63% (HR: 1.63; 95% CI: 1.52, 1.74; P for trend <0.001) higher risks of type 2 diabetes and ischemic heart disease, respectively, than those with serum remnant cholesterol of <0.4 mmol/L (Table 3). Similarly, participants with serum triglycerides of ≥ 2.3 mmol/L had 254% (HR: 3.54; 95% CI: 3.24, 3.87; P for trend <0.001) and 51% (HR: 1.51; 95% CI: 1.43, 1.59; P for trend <0.001) higher risks of type 2 diabetes and ischemic heart disease, respectively, than those with serum triglycerides of <0.4 mmol/L (Table 4). Besides, serum remnant cholesterol and triglycerides were also associated with higher risks of progression from ischemic heart disease to IHD-T2D multimorbidity. The HRs for the progression from ischemic heart disease to IHD-T2D multimorbidity were 2.14 (95% CI: 1.39, 3.31; P for trend <0.001) and 3.39 (95% CI: 2.25, 5.11; P for trend <0.001) for participants with serum remnant cholesterol of ≥1.0 mmol/L and triglycerides of ≥2.3 mmol/L, respectively (Tables 3 and 4). However, serum remnant cholesterol and triglycerides were not associated with the risks of progression from type 2 diabetes to IHD-T2D multimorbidity (Tables 3 and 4). Subgroup analyzes revealed that the associations between serum remnant cholesterol, triglycerides, and IHD-T2D multimorbidity did not differ significantly when stratified by lifestyle factors including diet, physical activity, and sleep pattern (Supplementary Data 4).

We next tested whether incorporating serum remnant cholesterol and triglycerides into the basal multivariate-adjusted regression models might increase the discrimination power for predicting cardiometabolic multimorbidity. Adding remnant cholesterol and triglycerides into the basal model substantially improves the discrimination performance with the C index increasing from 0.683 (95% CI: 0.655, 0.711) to 0.695 (95% CI: 0.668, 0.722; P for difference = 0.01) and from 0.676 (95% CI: 0.650, 0.702) to 0.700 (95% CI: 0.675, 0.725; P for difference = 2.45 × 10$^{-6}$), respectively, when considering the progression from ischemic heart disease to IHD-T2D multimorbidity (Supplementary Data 5). However, incorporating remnant cholesterol and triglycerides into the basal model did not markedly influence the discrimination performance when pooling the three cardiometabolic disease together (Supplementary Data 6).

### Mendelian randomization analysis

In this study, we derived a weighted genetic risks score comprising of 13 biologically relevant single nucleotide polymorphisms (SNPs), the candidate genes of which encode key enzymes and regulatory factors in triglyceride-rich lipoprotein metabolism pathway (Supplementary Data 7)[5]. Mendelian randomization analyzes revealed that genetically predicted remnant cholesterol and triglycerides were causally associated with higher risks of cardiometabolic multimorbidity. The causal odds ratios (ORs) for cardiometabolic multimorbidity, pooling all three cardiometabolic diseases together, and IHD-T2D multimorbidity per 1.0 mmol/L (0.98 standard deviation (SD)) increment in triglycerides were 1.21 (95% CI: 1.06, 1.39; P = 0.004) and 1.24 (95% CI: 1.06, 1.46;

**Table 1 | Baseline characteristics of participants stratified by serum remnant cholesterol and triglyceride categories**

| Variable | Serum remnant cholesterol, mmol/ L | | | | |
| --- | --- | --- | --- | --- | --- |
| | <0.4 | 0.4–<0.6 | 0.6–<0.8 | 0.8–<1.0 | ≥1.0 |
| Number of participants, n (%) | 41,529 (12.4) | 91,432 (27.4) | 95,887 (28.7) | 58,487 (17.5) | 46,695 (14.0) |
| Age, mean (SD), years | 52.5 (8.2) | 54.7 (8.2) | 56.1 (7.9) | 56.7 (7.8) | 56.2 (7.8) |
| Men, % | 29.3 | 36.7 | 43.7 | 48.7 | 55.7 |
| Townsend index, mean (SD) | −1.30 (3.08) | −1.38 (3.04) | −1.42 (3.02) | −1.44 (3.01) | −1.35 (3.05) |
| BMI, mean (SD), kg/m$^2$ | 24.7 (4.0) | 26.1 (4.4) | 27.3 (4.6) | 28.1 (4.5) | 28.8 (4.3) |
| Systolic blood pressure, mean (SD), mmHg | 130.6 (18.4) | 134.4 (18.4) | 137.8 (18.3) | 140.0 (18.2) | 142.0 (18.0) |
| Never-smokers, % | 61.1 | 59.4 | 57 | 55.2 | 51.9 |
| Access alcohol intake, % | 26.7 | 24.8 | 24.3 | 24.7 | 26.5 |
| Adequate physical activity, % | 75.5 | 73 | 70.4 | 68.3 | 66.4 |
| Diet quality score, mean (SD) | 3.5 (1.6) | 3.4 (1.6) | 3.3 (1.6) | 3.2 (1.6) | 3.0 (1.6) |
| Sleep quality score, mean (SD) | 3.2 (1.0) | 3.1 (1.0) | 3.0 (1.0) | 3.0 (1.0) | 2.9 (1.0) |
| Remnant cholesterol, mean (SD), mmol/L | 0.30 (0.09) | 0.51 (0.06) | 0.69 (0.06) | 0.89 (0.06) | 1.24 (0.28) |
| | Serum triglycerides, mmol/L | | | | |
| | <0.9 | 0.9–<1.3 | 1.3–<1.7 | 1.7–<2.3 | ≥ 2.3 |
| Number of participants, n (%) | 41,510 (11.4) | 91,395 (25.0) | 95,830 (26.2) | 58,458 (16.0) | 78,384 (21.4) |
| Age, mean (SD), years | 52.5 (8.2) | 54.7 (8.2) | 56.1 (7.9) | 56.7 (7.8) | 55.8 (7.9) |
| Men, % | 29.3 | 36.7 | 43.7 | 48.7 | 48.4 |
| Townsend index, mean (SD) | −1.30 (3.08) | −1.38 (3.04) | −1.42 (3.02) | −1.44 (3.01) | −1.36 (3.05) |
| BMI, mean (SD), kg/m$^2$ | 24.7 (4.0) | 26.1 (4.4) | 27.3 (4.6) | 28.1 (4.5) | 28.0 (4.5) |
| Systolic blood pressure, mean (SD), mmHg | 130.6 (18.4) | 134.4 (18.4) | 137.8 (18.3) | 140.0 (18.2) | 139.6 (18.5) |
| Never-smokers, % | 61.1 | 59.4 | 57 | 55.2 | 54.2 |
| Access alcohol intake, % | 26.7 | 24.8 | 24.3 | 24.7 | 25.8 |
| Adequate physical activity, % | 75.5 | 73 | 70.4 | 68.3 | 68.3 |
| Diet quality score, mean (SD) | 3.5 (1.6) | 3.4 (1.6) | 3.3 (1.6) | 3.2 (1.6) | 3.1 (1.6) |
| Sleep quality score, mean (SD) | 3.2 (1.0) | 3.1 (1.0) | 3.0 (1.0) | 3.0 (1.0) | 3.0 (1.0) |
| Triglycerides, mean (SD), mmol/L | 0.88 (0.31) | 1.16 (0.39) | 1.57 (0.53) | 2.10 (0.69) | 2.63 (1.43) |

*BMI* body mass index, *SD* standard deviation.

P = 0.007), respectively (Fig. 1). After rescaling the GRSs to represent the same magnitude (0.98 SD) of change with triglycerides, the corresponding causal ORs per 0.29 mmol/L increment in remnant cholesterol were 1.23 (95% CI: 1.07, 1.42; P = 0.005) and 1.26 (95% CI: 1.06, 1.50; P = 0.008), respectively. In consistency with the observational analysis, genetically determined blood remnant cholesterol and triglycerides were causally related to elevated risks of ischemic heart disease and type 2 diabetes but not stroke (Supplementary Fig. 2).

## Discussion

In this study of over 300,000 UK Biobank participants, serum remnant cholesterol and triglycerides were significantly associated with higher risks of cardiometabolic multimorbidity, particularly the risks of the progression from ischemic heart disease to IHD-T2D multimorbidity. Furthermore, Mendelian randomization analyzes conferred genetic evidence that higher levels of remnant cholesterol and triglycerides were causally related to higher risks of cardiometabolic multimorbidity, particularly in relation to IHD-T2D multimorbidity. The risks of IHD-T2D multimorbidity causally increased by 26% and 24% for each 1.0 mmol/L increment in triglycerides and 0.29 mmol/L increment in remnant cholesterol, respectively.

Previous studies have identified causal and positive associations between triglyceride-rich lipoproteins and atherosclerotic cardiovascular diseases[7,23]. Each 1.0 mmol/L increase in remnant cholesterol was causally associated with 2.8-fold higher risks of heart disease. Remnant cholesterol also reclassified up to 40% of incident myocardial infarction and ischemic heart disease beyond traditional cardiovascular risk factors among people free of cardiometabolic diseases[6]. In addition,

remnant cholesterol above 1.0 mmol/L and triglycerides above 2.0 mmol/L were linked to about two-fold higher cardiovascular mortality[24]. Besides, remnant cholesterol was associated with risks of developing diabetes after transplantation in renal transplant recipients[25]. Our results parallel these previous studies showing that serum remnant cholesterol and triglycerides were causally associated with higher risks of ischemic heart disease, type 2 diabetes, and cardiometabolic multimorbidity. We further demonstrated a markedly higher risk for progression from initial ischemic heart disease to IHD-T2D multimorbidity using a multistate model. In this regard, our results provide evidence endorsing the treatment of blood triglyceride-rich lipoproteins in people with existing cardiovascular diseases to mitigate the risks of multimorbidity progression.

In the present study, using biologically plausible genetic variants as the instrumental variables, we identified causal support for associations of remnant cholesterol and triglycerides with cardiometabolic multimorbidity. Drug target Mendelian randomization studies have demonstrated prospective evidence that the triglyceride-lowering alleles in the LPL pathway, which controls triglyceride hydrolysis, may be causally associated with lower risks of coronary heart disease and type 2 diabetes independent of LDL-C[26,27]. The recent REDUCE-IT trial suggested that high-dose omega-3 fatty acids, which inhibit the production but enhance the clearance of triglyceride-rich VLDL, were effective in reducing ischemic events in people with hypertriglyceridemia and prescribed with statins, albeit the results of this trial have been questioned due to the adverse effects of mineral oil used for the placebo comparator on blood lipids[4,28]. Hence, remnant cholesterol and triglycerides were not only biological determinants but also

**Table 2 | Observational associations between remnant cholesterol, triglycerides, and cardiometabolic multimorbidity when pooling type 2 diabetes, ischemic heart disease, and stroke together**

| | Serum remnant cholesterol, mmol/L | | | | | |
|---|---|---|---|---|---|---|
| | <0.4 | 0.4–<0.6 | 0.6–<0.8 | 0.8–<1.0 | ≥ 1.0 | P for trend |
| **Disease free to 1st CMD** | | | | | | |
| Cases/Total | 2216/41,526 | 6679/91,422 | 9438/95,868 | 7463/58,471 | 7426/46,676 | |
| Incidence | 44 | 60 | 82 | 108 | 137 | |
| HR, 95% CI | 1.00 | 1.06 (1.01, 1.11) | 1.17 (1.11, 1.22) | 1.36 (1.30, 1.43) | 1.59 (1.51, 1.67) | <1 × 10⁻²⁰ |
| **1st CMD to cardiometabolic multimorbidity** | | | | | | |
| Cases/Total | 170/2211 | 578/6679 | 914/9431 | 754/7460 | 809/7435 | |
| Incidence | 150 | 168 | 188 | 194 | 199 | |
| HR, 95% CI | 1.00 | 1.03 (0.86, 1.22) | 1.10 (0.93, 1.29) | 1.12 (0.95, 1.32) | 1.13 (0.96, 1.34) | 0.077 |
| | Serum triglycerides, mmol/L | | | | | |
| | <0.9 | 0.9–<1.3 | 1.3–<1.7 | 1.7–<2.3 | ≥ 2.3 | P for trend |
| **Disease free to 1st CMD** | | | | | | |
| Cases/Total | 3268/63,475 | 6670/91,913 | 6747/71,483 | 7869/66,304 | 11,663/72,329 | |
| Incidence | 42 | 60 | 79 | 100 | 139 | |
| HR, 95% CI | 1.00 | 1.09 (1.04, 1.13) | 1.20 (1.15, 1.25) | 1.33 (1.28, 1.39) | 1.66 (1.60, 1.73) | <1 × 10⁻²⁰ |
| **1st CMD to cardiometabolic multimorbidity** | | | | | | |
| Cases/Total | 195/3263 | 560/6669 | 618/6745 | 774/7856 | 1361/11,675 | |
| Incidence | 118 | 164 | 176 | 190 | 216 | |
| HR, 95% CI | 1.00 | 1.28 (1.09, 1.51) | 1.29 (1.10, 1.52) | 1.36 (1.16, 1.59) | 1.50 (1.29, 1.75) | 6.67 × 10⁻⁷ |

Incidence was calculated as cases per 10,000 person-years. Hazard ratios were calculated via multistate modeling adjusting for age, sex, body mass index, systolic blood pressure, smoking status, pack-years of smoking, alcohol intake, physical activity, diet quality score, sleep quality score, and fasting time.
*CI* confidence interval, *CMD* cardiometabolic disease, *HR* hazard ratio.
Statistical significance is set at two-sided *P* value < 0.05.

**Table 3 | Observational associations between remnant cholesterol and cardiometabolic multimorbidity when pooling type 2 diabetes and ischemic heart disease together**

| | Serum remnant cholesterol, mmol/L | | | | | |
|---|---|---|---|---|---|---|
| | <0.4 | 0.4–<0.6 | 0.6–<0.8 | 0.8–<1.0 | ≥ 1.0 | P for trend |
| **Disease free to T2D** | | | | | | |
| Cases/Total | 495/41,529 | 1807/91,432 | 2928/95,887 | 2511/58,487 | 2910/46,695 | |
| Incidence | 10 | 16 | 25 | 36 | 53 | |
| HR, 95% CI | 1.00 | 1.23 (1.11, 1.36) | 1.48 (1.35, 1.63) | 1.84 (1.67, 2.03) | 2.46 (2.23, 2.71) | <1 × 10⁻²⁰ |
| **Disease free to IHD** | | | | | | |
| Cases/Total | 1206/41,529 | 3604/91,432 | 5016/95,887 | 3988/58,487 | 3739/46,695 | |
| Incidence | 24 | 32 | 43 | 57 | 68 | |
| HR, 95% CI | 1.00 | 1.08 (1.01, 1.16) | 1.21 (1.14, 1.29) | 1.45 (1.36, 1.55) | 1.63 (1.52, 1.74) | <1 × 10⁻²⁰ |
| **T2D to IHD-T2D multimorbidity** | | | | | | |
| Cases/Total | 34/469 | 114/1720 | 240/2810 | 175/2392 | 222/2766 | |
| Incidence | 149 | 133 | 173 | 147 | 149 | |
| HR, 95% CI | 1.00 | 0.84 (0.57, 1.24) | 1.09 (0.76, 1.56) | 0.93 (0.64, 1.34) | 0.97 (0.67, 1.39) | 0.416 |
| **IHD to IHD-T2D multimorbidity** | | | | | | |
| Cases/Total | 23/1202 | 120/3601 | 189/5000 | 196/3974 | 220/3732 | |
| Incidence | 35 | 61 | 69 | 89 | 101 | |
| HR, 95% CI | 1.00 | 1.54 (0.98, 2.41) | 1.57 (1.02, 2.43) | 2.01 (1.30, 3.10) | 2.14 (1.39, 3.31) | 8.89 × 10⁻⁵ |

Incidence was calculated as cases per 10,000 person-years. Hazard ratios were calculated via multistate modeling adjusting for age, sex, body mass index, systolic blood pressure, smoking status, pack-years of smoking, alcohol intake, physical activity, diet quality score, sleep quality score, and fasting time.
*CI* confidence interval, *HR* hazard ratio, *IHD* ischemic heart disease, *T2D* type 2 diabetes.
Statistical significance is set at two-sided *P* value < 0.05.

promising therapeutic targets for the treatment of cardiometabolic diseases and prevention of progression from single disease to cardiometabolic multimorbidity.

Direct assay of triglyceride-rich lipoproteins requires labor-intensive protocols like ultracentrifugation or immunoseparation, which might hinder its implementation in large-scale cohort studies like the UK Biobank[29]. In contrast, in the present study, we used remnant cholesterol and triglycerides as two simple and straightforward surrogates for triglyceride-rich lipoproteins. Remnant cholesterol and triglycerides were strongly correlated but not identical.

**Table 4 | Observational associations between triglycerides and cardiometabolic multimorbidity when pooling type 2 diabetes and ischemic heart disease together**

| | Serum triglycerides, mmol/L | | | | | |
|---|---|---|---|---|---|---|
| | **<0.9** | **0.9–<1.3** | **1.3–<1.7** | **1.7–<2.3** | **≥ 2.3** | **P for trend** |
| Disease free to T2D | | | | | | |
| Cases/Total | 582/63,480 | 1508/91,927 | 1863/71,493 | 2652/66,318 | 4986/72,359 | |
| Incidence | 7 | 13 | 22 | 33 | 59 | |
| HR, 95% CI | 1.00 | 1.33 (1.21, 1.47) | 1.74 (1.59, 1.91) | 2.26 (2.07, 2.48) | 3.54 (3.24, 3.87) | $<1 \times 10^{-20}$ |
| Disease free to IHD | | | | | | |
| Cases/Total | 1871/63,480 | 3852/91,927 | 3760/71,493 | 4141/66,318 | 5517/72,359 | |
| Incidence | 24 | 34 | 43 | 52 | 65 | |
| HR, 95% CI | 1.00 | 1.13 (1.07, 1.19) | 1.23 (1.17, 1.30) | 1.32 (1.25, 1.40) | 1.51 (1.43, 1.59) | $<1 \times 10^{-20}$ |
| T2D to IHD-T2D multimorbidity | | | | | | |
| Cases/Total | 27/546 | 102/1426 | 137/1777 | 198/2536 | 372/4760 | |
| Incidence | 110 | 147 | 158 | 157 | 146 | |
| HR, 95% CI | 1.00 | 1.21 (0.79, 1.85) | 1.27 (0.84, 1.92) | 1.24 (0.83, 1.86) | 1.15 (0.77, 1.71) | 0.500 |
| IHD to IHD-T2D multimorbidity | | | | | | |
| Cases/Total | 25/1865 | 100/3845 | 142/3753 | 177/4123 | 378/5505 | |
| Incidence | 25 | 47 | 68 | 78 | 120 | |
| HR, 95% CI | 1.00 | 1.65 (1.07, 2.57) | 2.05 (1.34, 3.14) | 2.39 (1.57, 3.64) | 3.39 (2.25, 5.11) | $4.66 \times 10^{-15}$ |

Incidence was calculated as cases per 10,000 person-years. Hazard ratios were calculated via multistate modeling adjusting for age, sex, body mass index, systolic blood pressure, smoking status, pack-years of smoking, alcohol intake, physical activity, diet quality score, sleep quality score, and fasting time.

*CI* confidence interval, *HR* hazard ratio, *IHD* ischemic heart disease, *T2D* type 2 diabetes.

Statistical significance is set at two-sided *P* value < 0.05.

| Outcome | | OR, 95% CI | P |
|---|---|---|---|
| **Remnant cholesterol** | | | |
| First cardiometabolic disease | | 1.24 (1.16, 1.33) | $2.24 \times 10^{-9}$ |
| Cardiometabolic multimorbidity | | 1.23 (1.07, 1.42) | 0.005 |
| IHD−T2D multimorbidity | | 1.26 (1.06, 1.50) | 0.008 |
| **Triglycerides** | | | |
| First cardiometabolic disease | | 1.22 (1.15, 1.31) | $1.43 \times 10^{-9}$ |
| Cardiometabolic multimorbidity | | 1.21 (1.06, 1.39) | 0.004 |
| IHD−T2D multimorbidity | | 1.24 (1.06, 1.46) | 0.007 |

**Fig. 1 | Genetic associations between remnant cholesterol, triglycerides, and cardiometabolic multimorbidity via one-sample Mendelian randomization among the UK Biobank participants (*n* = 376,712 for remnant cholesterol and *n* = 411,930 for triglycerides).** Causal odds ratios were calculated adjusting for age and sex. Data are presented as odd ratios with 95% confidence intervals. The dot refers to the odd ratio while the solid line refers to the 95% confidence interval. The dash line refers to odd ratio value of one as the reference. Abbreviations: CI confidence interval, IHD ischemic heart disease, OR odds ratio, T2D type 2 diabetes. Statistical significance is set at two-sided *P* value < 0.05.

Given that there have been varied definitions for the remnants of triglyceride-rich lipoproteins, the term remnant cholesterol hereby refers to all cholesterol contents in triglyceride-rich lipoproteins, including chylomicron remnants, VLDLs, and intermediate-density lipoproteins[30,31]. In the present study, about 14% and 21 % of the UK Biobank participants had nonfasting serum remnant cholesterol above 1.0 mmol/L and triglycerides above 2.3 mmol/L, respectively. The distributions of remnant cholesterol and triglycerides in the UK Biobank were comparable with those in the Copenhagen General Population Study[1,5]. In the latter study, about 27% and 21% of participants had mild-to-moderate elevated nonfasting triglycerides and remnant cholesterol above 2.0 mmol/L and 1.0 mmol/L, respectively[1,5]. Random nonfasting triglycerides were, on average, 20-25% higher than the fasting values and reached the bottom after an overnight fast[32]. However, postprandial remnant cholesterol metabolism remains to be established, and a clinically meaningful cutoff value of remnant cholesterol is still lacking, which could facilitate translating our results into clinical practice.

Notwithstanding we showed a causal link between triglyceride-rich lipoproteins and type 2 diabetes risk in this study, elevated remnant cholesterol and triglycerides were unlikely to be mechanistically involved in the pathogenesis of type 2 diabetes. Seemingly, the genetic associations between remnant cholesterol, triglycerides, and type 2 diabetes could be attributed to increased likelihood of diabetes detection and diagnosis in participants with elevated triglycerides as clinicians often use elevated triglycerides as a biomarker of insulin resistance[33]. We also showed that baseline serum remnant cholesterol and triglycerides were associated with higher risks of progression from ischemic heart disease to IHD-T2D multimorbidity but not the progression from type 2 diabetes to IHD-T2D multimorbidity, suggesting differential roles of triglyceride-rich lipoproteins in disease progression among patients that developed heart disease or diabetes.

Mechanisms underlying the causal associations between triglyceride-rich lipoproteins and cardiometabolic multimorbidity may involve their atherogenic and pro-inflammatory properties[1,15,16]. Besides, most people with elevated triglycerides or remnant cholesterol are accompanied by overweight, hyperglycemia, or fatty liver[15]. Insulin resistance could impair hepatic clearance of VLDL, leading to overaccumulation of triglyceride-rich lipoprotein remnants[34]. Lipid dysmetabolism in type 2 diabetes is characterized by hypertriglyceridemia, normal or slightly higher LDL-C, and increased apolipoprotein B secretion, the main structural components of triglyceride-rich lipoproteins[15]. Future studies are warranted to disentangle the roles and related mechanisms of remnant cholesterol and triglycerides in the progression of cardiometabolic disease to multimorbidity.

Strengths of this study include the large sample size and comprehensive case identification using primary care, death registry, and hospital inpatient records, which allowed us to develop a multistate disease transition model with sufficient statistical power. Moreover, a combination of multistate modeling and Mendelian randomization could, to a greater extent, address the residual confounding and reverse causality, which is major methodological issues in

observational studies. In addition, remnant cholesterol and triglycerides were measured from the nonfasting blood samples, which could better characterize lipid profiles over a 24 h cycle in a real-world setting[5,16]. Nonfasting triglycerides also had a better diagnostic performance for cardiovascular diseases than the fasting values[5,32].

This study also has several limitations. First, blood samples were collected only once at baseline and thus were unable to capture long-term changes before and after the incidence of first cardiometabolic diseases. Although estimates from Mendelian randomization could indicate the life course effects of remnant cholesterol and triglycerides, repeated measurements could help to understand the impact of triglyceride-rich lipoproteins on the progression to cardiometabolic multimorbidity. Second, information concerning lipid-lowering medication use was also collected at baseline. It is unclear whether changes in the use of lipid-lowering medications after the incidence of first cardiometabolic disease could influence the risks of progression to multimorbidity via triglyceride-rich lipoproteins. Third, the potential pleiotropy of genetic instruments might lead to biased risk estimates for any Mendelian randomization analysis. In this study, we only included biologically plausible genetic variants involved in triglyceride-rich lipoproteins, minimizing the risk of pleiotropic effects. Fourth, the UK Biobank participants tended to have a better socioeconomic status than the entire UK population. However, valid findings of the exposure-outcomes associations do not necessarily require the participants to be representative of the general population at large[35]. Fifth, although the Harrell's C index significantly increased after adding remnant cholesterol and triglycerides into the basal model, a risk prediction model specific for developing cardiometabolic morbidity is still lacking. Also, it remains to be studied whether remnant cholesterol and triglycerides could improve the discrimination and calibration power of the risk prediction model for progression from first cardiometabolic disease to multimorbidity. Sixth, cardiometabolic health status could influence clinical decision to prescribe lipid-lowering medication, which might a collider on the pathway from remnant cholesterol and triglycerides to cardiometabolic risk, leading to biased risk estimates[36,37]. However, sensitivity analysis including participants receiving lipid-lowering medication at the second-stage regression yielded similar results compared to the main analysis.

In conclusion, remnant cholesterol and triglycerides were both observationally and genetically linked to higher cardiometabolic multimorbidity risks. These findings imply that triglyceride-rich lipoproteins could serve potential risk factors and therapeutic targets for the prevention and treatment of cardiometabolic diseases, which necessitate further validation through randomized controlled trials focusing on triglyceride-rich lipoproteins.

## Methods

### Study population
The UK Biobank has obtained ethical approval from the North West Multi-centre Research Ethics Committee as a Research Tissue Bank approval (Approval Number: 11/NW/0382, https://www.ukbiobank.ac.uk/learn-more-about-uk-biobank/about-us/ethics). Besides, we also got approval for performing data analysis from Peking University Third Hospital Medical Science Research Ethics Committee (Approval Number: S2024080). All UK Biobank participants provided written informed consent for their participation in the study and access to their medical records. The present study was performed under Application ID 44430. The UK Biobank is a prospective study of over 500,000 UK adults aged between 37 and 73 years that were recruited between 2006 and 2010[38]. During the baseline visit to the local assessment centers, participants were invited to complete a nurse-administered questionnaire collecting data on demographic, socioeconomic, lifestyle, and medical history, had physical measurements recorded and provided biological samples for biochemical and genetic analyzes. In the observational analysis, participants were excluded if

they had any cardiometabolic diseases (type 2 diabetes, ischemic heart disease, and stroke) at baseline, were prescribed lipid-lowering medications ($n = 98,828$), or had missing values for remnant cholesterol ($n = 73,390$) or triglycerides ($n = 33,284$). Therefore, 334,030 (men% = 42.5%) and 365,577 (men% = 42.1%) eligible participants were included in the observational analysis of remnant cholesterol and triglycerides, respectively (Supplementary Fig. 3a). In the Mendelian randomization analysis, participants were excluded if they were not white British ($n = 59,895$), did not pass the quality control for genotyping ($n = 3636$), or had missing values for remnant cholesterol ($n = 73,390$) or triglycerides ($n = 33,284$). To avoid collider bias, we included participants that used lipid-lowering medications in both first and second stage of Mendelian randomization analysis. Therefore, 376,712 (men% = 46.2%) and 411,930 (men% = 45.9%) eligible participants were included in the Mendelian randomization analysis of remnant cholesterol and triglycerides, respectively (Supplementary Fig. 3b).

### Serum remnant cholesterol and triglyceride measurements
Remnant cholesterol and triglycerides are two simple measurements of triglyceride-rich lipoproteins[1]. Nonfasting blood samples were collected during the baseline examination[39]. Serum total cholesterol and triglyceride concentrations were measured using an enzymatic colorimetric method on Beckman Coulter AU5800 analyzers. Serum LDL-C concentrations were directly measured using an enzymatic protective selection method on Beckman Coulter AU5800 analyzers. Serum HDL-C concentrations were measured using the enzyme immune-inhibition analysis (Beckman Coulter AU5800). Serum remnant cholesterol was calculated as total cholesterol minus HDL-C minus LDL-C[5,6,30]. A circulating triglyceride above 1.7 mmol/L (150 mg/dL) is a widely accepted cutoff point for hypertriglyceridemia[2]. Therefore, we categorized triglycerides into five groups: <0.9 mmol/L (80 mg/dL), 0.9−<1.3 mmol/L (80− <115 mg/dL), 1.3−<1.7 mmol/L (115 − <150 mg/dL), 1.7−<2.3 mmol/L (150− <204 mg/dL), and ≥ 2.3 mmol/L (≥204 mg/dL). However, there is no recommended cutoff point for elevated blood remnant cholesterol yet. We arbitrarily categorized remnant cholesterol into five groups: <0.4 mmol/L (15 mg/dL), 0.4 - <0.6 mmol/L (15− < 23 mg/dL), 0.6−0.8 mmol/L (23 - < 31 mg/dL), 0.8−<1.0 mmol/L (31− < 39 mg/dL), and ≥ 1.0 mmol/L (≥39 mg/dL), yielding a similar pattern of participant distribution in each category between remnant cholesterol and triglycerides (Table 1).

### Genetic instruments of remnant cholesterol and triglycerides
Genotyping was performed using the UK BiLEVE array and the UK Biobank Axiom array (similarity >95%)[40]. Mendelian randomization has three core instrumental variable assumptions: 1) the genetic instruments are strongly associated with the exposure; 2) the genetic instruments are not associated with confounding factors; 3) the genetic instruments are exclusively associated with the outcome via the exposure. To avoid pleiotropy of genetic instruments, we retrieved the 16 gain-of-function and loss-of-function single nucleotide polymorphisms in a previous study by Kaltoft et al.[5]. Candidate genes of these SNPs encode key enzymes, regulatory factors, and lipoproteins that have well-established effects on triglyceride-rich lipoprotein metabolism, including *LPL*, angiopoietin-like 3 (*ANGPTL3*), angiopoietin-like 4 (*ANGPTL4*), apolipoprotein C-III (*APOC3*), and apolipoprotein A-V (*APOAV*)[7,41,42] Three rare variants (rs569107562, rs749131121, and rs267606655) were not available in the UK Biobank study. Such genetic instrument selection strategy was comparable to that used in drug target Mendelian randomization analysis, which could reduce the possibility of horizontal pleiotropy and weak instrument bias[43]. A total of 13 biologically relevant SNPs were included as genetic instruments for remnant cholesterol and triglycerides (Supplementary Data 7). We calculated weighted GRSs as aggregated genetic instrumentals for blood remnant cholesterol and triglycerides. The weighted GRS was calculated by summing the products of the

number of lipid-raising alleles and the effect size of each SNP on blood remnant cholesterol and triglycerides (Supplementary Data 7). The weighted GRSs could explain around 2.95% and 3.01% variance of serum remnant cholesterol and triglycerides, respectively (Supplementary Data 7).

## Cardiometabolic multimorbidity

The prevalent and newly on-set cardiometabolic diseases, namely type 2 diabetes, ischemic heart disease, and stroke, were identified via linkage to primary care records, hospital inpatient records, and death registry records and outcomes were coded using the International Classification of Disease, 10th revision codes: type 2 diabetes, E11; ischemic heart disease, I21−I25; stroke, I60-I64. We censored participants at the end of follow-up, at the date of the first occurrence of any cardiometabolic disease or development of multimorbidity, and at the date of loss to follow-up, whichever was first. Cardiometabolic multimorbidity was defined as the concurrence of at least two cardiometabolic diseases, like the co-existence of ischemic heart disease and type 2 diabetes (IHD-T2D multimorbidity) and the co-existence of stroke and ischemic heart disease (IHD-stroke multimorbidity). In the observational analysis, we only included participants that were free of any cardiometabolic diseases at baseline and investigated the associations between triglyceride-rich lipoproteins, newly onset cardiometabolic diseases, and subsequent progression to cardiometabolic multimorbidity (Supplementary Fig. 1). In the Mendelian analysis, we also included participants with existing cardiometabolic diseases and cardiometabolic multimorbidity at baseline.

## Covariates

Age and sex were self-reported. Townsend deprivation index is an indicator of socioeconomic status, which was assessed based on local percentages of unemployment, non-car ownership, non-home ownership, and household overcrowding. A higher Townsend index suggests a more disadvantaged socioeconomic status. Body mass index was calculated as body weight divided by the square of height and expressed as kg/m². Systolic blood pressure was measured by a digital blood pressure monitor or a manual sphygmomanometer. Smoking status was categorized as never, former, and current smokers. Pack-years of smoking were calculated as the number of cigarettes smoked per day divided by twenty, multiplied by the number of years of smoking. Alcohol intake was assessed by a food frequency questionnaire concerning beverage types and frequency. Low alcohol intake was defined as <22 units/day in men and <15 units/day in women[44]. Physical activity was assessed by the International Physical Activity Questionnaire short form. Adequate physical activity was defined as ≥150 min/week of moderate activity, or ≥75 min/week of vigorous activity, or ≥150 min/week of combined moderate and vigorous activity, ≥5 times/week of moderate activity, or ≥1 time week of vigorous activity[45].

We derived a diet quality score as the indicator of diet quality, which is based on intakes of fruits, vegetables, processed meat, red meat, fish, whole grains, and refined grains[46]. Diet was assessed via a food frequency questionnaire at baseline. One point was assigned to each food item if: the average daily intake of: 1) fruits ≥ 3 servings/day; 2) vegetables ≥ 3 servings/day; 3) fish ≥ 2 servings/week; 4) processed meat ≤ 1 servings/week; 5) red meat ≤ 1.5 servings/week; 6) whole grain ≥ 3 servings/week; 7) refined grain ≤ 1.5 servings/week. The diet quality score ranged from 0 to 7 with a higher score indicating a better diet quality. A higher diet quality score indicated better adherence to a healthy dietary pattern. In addition, we also constructed a sleep quality score as an indicator of healthy sleep patterns, which is based on chronotype, sleep duration, frequency of insomnia, snoring, and daytime sleepiness[46]. Sleep pattern was evaluated via a touchscreen questionnaire during the baseline assessment. One point was assigned to each item if: 1) early chronotype; 2) sleep 7 to 8 h per day; 3) never or

rarely had insomnia symptoms; 4) no self-reported snoring; 5) no frequent daytime sleepiness. Therefore, the sleep quality score ranged from 0 to 5 with a higher score indicating healthier sleep pattern.

## Lipid-lowering medication

During the baseline assessment, the participants were invited to take a verbal interview conducted by trained staff completed at the assessment center. Triglyceride-lowering medication was identified using the following treatment/medication code: 1140861954, fenofibrate; 1140861924, bezafibrate; 1141157260, bezafibrate product; 1140861944, clofibrate; 1140862026, ciprofibrate; 1140910670 niacin; 1140861868, nicotinic acid product; 1193, omega-3/fish oil supplement. Cholesterol-lowering medication was identified using the following codes: 1141146234, atorvastatin; 1140888594, Fluvastatin; 1140888648, pravastatin; 1141192410, rosuvastatin; 1140861958, simvastatin. In addition, the participants were asked whether they regularly took cholesterol-lowering medications via a touchscreen questionnaire during the baseline assessment.

## Statistical analysis

**Multistate modeling.** In the observational analysis, we constructed multistate models to assess the associations of blood remnant cholesterol and triglycerides with incident cardiometabolic diseases and the progression of the first cardiometabolic disease to cardiometabolic multimorbidity. The multistate model, as an extension of competing risk model, was increasingly used to study complex disease progression[47]. In this study, we defined three states of participants, namely free of cardiometabolic disease, incidence of first cardiometabolic disease, and development of cardiometabolic morbidity, allowing two disease transitions, namely 1) from baseline to first cardiometabolic disease and 2) from first cardiometabolic disease to multimorbidity (Supplementary Fig. 1a). For participants with the same date for first cardiometabolic disease and multimorbidity, the date of first cardiometabolic disease was calculated as the date of multimorbidity minus the median times for the transition from incident first cardiometabolic disease to multimorbidity[48]. In addition, we also constructed a multistate model that also included ischemic heart disease and type 2 diabetes allowing four transitions, namely 1) from baseline to incident type 2 diabetes, 2) from baseline to ischemic heart disease, 3) from type 2 diabetes to IHD-T2D multimorbidity, and 4) from ischemic heart disease to IHD-T2D multimorbidity. HRs for disease transitions across different levels of serum remnant cholesterol and triglycerides were calculated using the flexible parametric survival model, which allows time-dependent effects during disease transitions[49,50]. The flexible parametric survival model used restricted cubic splines to estimate hazard functions. The number of knots was selected based on Akaike's Information Criteria and Bayesian Information Criteria. Age was used as the time scale. The proportional hazards assumption was tested using Schoenfeld residuals via transition-specific Cox regression. Age was treated as a time-varying variable. The multivariate models were adjusted for age, sex, Townsend index, BMI, systolic blood pressure, smoking status, pack-years of smoking, alcohol intake, physical activity, diet quality score, sleep quality score, and fasting time. Tests for trends were conducted by treating the median value of each lipid category as continuous variables. The discrimination performance of models with and without remnant cholesterol or triglycerides (the basal model) was compared using Harrell's C index.

**Mendelian randomization.** We used one-sample Mendelian randomization to study the causal associations of remnant cholesterol and triglycerides with incident cardiometabolic disease and multimorbidity. The Causal ORs were calculated via the two-stage least squares method[51]. For comparing remnant cholesterol with triglycerides, the GRSs were rescaled to 1.0 mmol/L increment in triglycerides

(standard deviation (SD): 1.02 mmol/L; 0.98 SD) and 0.29 mmol/L increment in remnant cholesterol (SD: 0.30 mmol/L; 0.98 SD). Because genotype is determined at conception and is less susceptible to confounding, the model only adjusted for age and sex. In the sensitivity analysis, we included participants receiving lipid-medications in the second-stage regression in Mendelian randomization to test whether use of lipid-lowering medication might be a collider on the pathway from remnant cholesterol and triglycerides to cardiometabolic multimorbidity. The endpoints of interest in the Mendelian randomization analysis were cardiometabolic multimorbidity.

Statistical analysis was performed in Stata/MP version 17.0 and R version 4.2.2 with significance set at two-sided $P < 0.05$.

### Reporting summary
Further information on research design is available in the Nature Portfolio Reporting Summary linked to this article.

## Data availability
This research was conducted using the UK Biobank Resource (https://www.ukbiobank.ac.uk/) under Application ID 44430. Data from UK Biobank is accessible to eligible researchers via applying to www.ukbiobank.ac.uk. Data supporting the findings of this study are available in the article and its Supplementary information. Source data are provided with this paper.

## Code availability
Multistate model was developed using the stmerlin stata command: https://github.com/RedDoorAnalytics/stmerlin. Forest plots were developed using the R package: https://cran.r-project.org/web/packages/forestploter. Analytical codes have been deposited at Zenodo (https://doi.org/10.5281/zenodo.10560862)[52].

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

## Acknowledgements

This work was supported by the National Natural Science Foundation of China (grant numbers: 82204026 (receiver: Y.Z.) and 82173499 (receiver: T.H.)).

## Author contributions

Y.Z. and T.H. conceptualized this study. Y.Z., Z.Z., Y.L., W.X., Z.S., N.H., W.W. and X.D. carried out the data analysis. Y.Z. carried out data visualization and drafted the manuscript. T.H. and R.C. reviewed and edited the manuscript.

## Competing interests

The authors declare no competing interests.
