## [Peer Review File · Nature Communications]

REVIEWER COMMENTS

Reviewer #1 (Remarks to the Author):

Zhao et al. have done an interesting and extensive observational and genetic study investigating remnant cholesterol and triglycerides as risk factors for cardiometabolic disease, which is defined as ischemic heart disease, stroke, and diabetes type 2. Remnant cholesterol was calculated from total cholesterol minus HDL cholesterol minus LDL cholesterol (measured with a direct assay). The authors selected genetic variants with known biological effects on triglyceride-rich lipoprotein metabolism to avoid pleiotropy, and added other variants located close to the genomic position of genes involved in triglyceride-rich lipoprotein metabolism. The authors then did observational analyses in which they predicted the risk of cardiometabolic morbidity and progression from a first cardiometabolic disease event to multimorbidity, as well as genetic analyses to investigate if there is evidence that these relationships are causal. They additionally used novel non-linear Mendelian randomization (MR).

The manuscript is well presented and reproducible, the choice of genetic variants limits pleiotropy and using remnant cholesterol calculated from directly measured LDL adds some novelty. Furthermore, the study of progression from a first cardiometabolic disease event to multimorbidity appears novel.

Major comments:

Elevated remnant cholesterol/triglycerides are probably not mechanistically involved in the pathogenesis of diabetes, as elevated plasma triglycerides do not increase fasting insulin or fasting glucose (PMID: 21282362). This means that a likely explanation for the genetic association between elevated remnant cholesterol/triglycerides and diabetes likely is due to increased likelihood of diabetes detection and diagnosis in individuals with elevated triglycerides, as clinicians often use elevated triglycerides as a marker indicative of insulin resistance; thus, the genetic causal association does not have any implications for prevention of diabetes. This needs to be mentioned in the Discussion section.

Indeed, pooling of ischemic heart disease, stroke, and diabetes is difficult to make sense of in genetic causal analyses. Therefore, please also present sensitivity analyses only pooling ischemic heart disease and stroke, which from a causal perspective would make much more sense as elevated remnant cholesterol/triglycerides are causal risk factors for both these diseases. This analyses preferably should also be presented in the main part of the paper but could also be presented in the supplementary section.

Introduction: "Cardiometabolic multimorbidity" should be defined. The aim of the article does not seem to be to "compare" observational and genetic analyses, but rather to test if elevated remnant cholesterol and triglycerides are associated with increased risk of cardiometabolic multimorbidity (using observational and genetic analyses). The aim should be rephrased to harmonize with the results and conclusion.

It is not completely clear if users of statins were included or excluded from the various analyses performed (please mention that where relevant in the paper). As high risk of cardiovascular disease and diabetes status influences the clinical decision to prescribe statins, statin use is a likely collider on the

pathway from remnant cholesterol to cardiometabolic risk. Therefore, (if you have not already done it) please perform sensitivity analyses where individuals on statins are excluded in observational analyses and in the first stage regression in genetic analyses, but not excluded from the second-stage regression in Mendelian randomization. These sensitivity analyses can be presented in the Supplementary, with the findings here from summarized in the Results section of the main paper. Please also mention in the limitations section of the Discussion, that statin use is a likely collider on the pathway from remnant cholesterol to cardiometabolic risk, and what that means for the present results.

Results: Separate analyses of the endpoints included in the composite endpoint (ischemic heart disease, stroke, and diabetes) should be done, otherwise it cannot be distinguished if the results are all driven by an association with ischemic heart disease (which is already well known), an association with stroke, and/or an association with diabetes (which as mentioned above in contrast to ischemic heart disease and stroke, likely is not mechanistic, but diagnostic).

Results: Non-linear Mendelian randomization is a method which currently (if it is used) may not be advisable, and therefore should be used with extreme caution. As exemplified with the article on Vitamin D levels (PMID: 34717822), there are risks of unexpected biases (PMID: 36528346); tools for addressing these have not yet been developed. In the present paper, non-linear Mendelian randomization seems somewhat redundant and could be dropped; otherwise, extensive sensitivity analyses are required, including using the doubly-ranked method (PMID: 36528346) and estimation of associations between the genetic risk score and measured confounders in each of the strata of exposure residuals.

Minor comments:

Methods: Were the same individuals included in both observational and genetic analyses?

Abstract: "Cardiometabolic disease event", and "multimorbidity" should be defined in Methods. The event of interest in the Mendelian randomization analyses should also be specified.

Reviewer #2 (Remarks to the Author):

The manuscript by Zhao Y, et al. studies the associations between elevated remnant cholesterol and triglycerides levels with the risk of developing cardiometabolic multimorbidity from the UK Biobank encompassing over 370,000 individuals. The results suggest that high levels of these two markers are associated with an increased risk of developing multiple cardiometabolic diseases. In particular, triglyceride-rich lipoproteins could be both potential risk factors and therapeutic targets for the prevention and treatment of cardiometabolic diseases.

The manuscript is overall well-written and organized, however, several points should be better discussed and clarified.

- Did the study account for any lifestyle or environmental factors that could influence the progression of cardiometabolic diseases to multimorbidity?

- How was the progression from a first cardiometabolic disease event to multimorbidity defined? Were there any specific criteria or time intervals used to identify multimorbidity?

- The abstract reports "a median follow-up of 12.6 years", but what is the span? Can the results be affected if adjusting for the years passed between baseline and follow-up?

- While the introduction briefly mentions previous studies and their findings regarding the association of triglycerides with cardiovascular disease, it would benefit from a more comprehensive review of the existing literature, including any conflicting or inconclusive results. For instance:

<https://doi.org/10.1016/j.atherosclerosis.2022.06.205>, <https://doi.org/10.3389/fgene.2023.1117778>, <https://doi.org/10.1016/j.numecd.2011.06.003>, <https://doi.org/10.3390/nu13061935>

- What are the potential implications for clinical practice, and what further research is needed to evaluate the effectiveness and feasibility of interventions aimed at reducing triglyceride-rich lipoproteins? This should probably be better discussed.

- Since alcohol was estimated from food frequency questionnaires, why were no other dietary factors considered in the analysis?

- The study adjusted for many covariates, but it did not incorporate dietary information than alcohol. Triglyceride levels can be heavily influenced by diet and lifestyle factors, such as the consumption of high-fat and high-sugar foods. What could be the impact of dietary information on the results?

- The authors in the "Discussion" mention: "Besides, remnant cholesterol was associated with risks of developing diabetes after transplantation in renal transplant recipients. Our results were consistent with these findings", but this is not clear how the presented results are supporting this statement, please clarify.

- How were the instrumental variables for the Mendelian randomization analysis selected? Were they associated with the exposure only or also independent of potential confounders?

Reviewer #3 (Remarks to the Author):

With interest I've read the paper Elevated blood remnant cholesterol and triglycerides are causally related to the risks of cardiometabolic multimorbidity, by Zhao et al. The authors aimed to assess the observational and causal link between remnant cholesterol and triglycerides with the risks of cardiometabolic multimorbidity. Cardiometabolic morbidity was defined as the concomitant prevalence of two+ cardiometabolic diseases such as T2D, IHD and stroke. The authors performed analyses in a subset of the UK Biobank. The authors only investigated incident cases, and excluded participants with prior cardiometabolic disease. In 371,234 participants, they quantified a total of 39,805 incident cases of cardiometabolic disease, fo which 15,547 progressed to cardiometabolic multimorbidity. Using these data and by employing various models, the authors showed that remnant cholesterol and triglycerides both were associated with increased risk of incident cardiometabolic disease and with progression to multimorbidity. They also showed that these associations seem causal, as quantified by both linear and nonlinear Mendelian Randomization.

Although these results are intuitive, I commend the authors for quantifying these associations. There are a few comments and questions, especially with regards to MR:

Methods:

1. GRS: internally weighted, this does increase the risk for weak instrument bias (also see 2). For TG at least, there is the option of validating results using externally weighed GWAS values. Suggest elaborating on this limitation and performing sensitivity analyses with externally weighed SNPs. (e.g. using the weights from Kaltoft et al).

2. Internally weighted, some instruments are invalid (as P-value for some of the instruments $> 5e-4$). Suggest performing MR-egger and perform analyses where weak SNPs are excluded.

3. Current MR studies are advised to follow the STROBE-MR guidelines and include the checklist in their supplementals.

5. The use of ICD-10 codes ensures that the outcome is relatively clean from bias. However, The inclusion of I20 might be a bit problematic, as this refers to Angina pectoris, including many non-specified. This increases the risk of including patients with AP symptoms that refer to something different than purely cardiac. Please consider performing sensitivity analyses with only 'hard' coronary artery disease (e.g. excluding I20).

6. Potentially interesting: Progression from which to what? E.g. DM2 to ischemic heart disease? Suggest splitting it out.

7. The authors excluded participants with cardiovascular disease before inclusion date, which could introduce participation bias in their MR study. (MR is lifetime risk on CVD, so excluding participants based on disease before inclusion in the biobank will likely result in skewed results). How does this affect your results and how large is this issue? Would reanalysis with participants with CVD before inclusion alter the results?

Results:

1. Harrell's C index did significantly increase, but does numerically not add any value. Suggest rewriting as the added clinical value is extremely limited (or even absent).

Discussion:

1. The discussion is rather underwhelming. Suggest expanding upon the novelties of this study, and expand on non-linear relationships for example.

Minor comments:

- Some minor language errors (line 78 for example, 349-352)

REVIEWER COMMENTS

Reviewer #1 (Remarks to the Author):

Zhao et al. have done an interesting and extensive observational and genetic study investigating remnant cholesterol and triglycerides as risk factors for cardiometabolic disease, which is defined as ischemic heart disease, stroke, and diabetes type 2. Remnant cholesterol was calculated from total cholesterol minus HDL cholesterol minus LDL cholesterol (measured with a direct assay). The authors selected genetic variants with known biological effects on triglyceride-rich lipoprotein metabolism to avoid pleiotropy, and added other variants located close to the genomic position of genes involved in triglyceride-rich lipoprotein metabolism. The authors then did observational analyses in which they predicted the risk of cardiometabolic morbidity and progression from a first cardiometabolic disease event to multimorbidity, as well as genetic analyses to investigate if there is evidence that these relationships are causal. They additionally used novel non-linear Mendelian randomization (MR). The manuscript is well presented and reproducible, the choice of genetic variants limits pleiotropy and using remnant cholesterol calculated from directly measured LDL adds some novelty. Furthermore, the study of progression from a first cardiometabolic disease event to multimorbidity appears novel.

Major comments:

Elevated remnant cholesterol/triglycerides are probably not mechanistically involved in the pathogenesis of diabetes, as elevated plasma triglycerides do not increase fasting insulin or fasting glucose (PMID: 21282362). This means that a likely explanation for the genetic association between elevated remnant cholesterol/triglycerides and diabetes likely is due to increased likelihood of diabetes detection and diagnosis in individuals with elevated triglycerides, as clinicians often use elevated triglycerides as a marker indicative of insulin resistance; thus, the genetic causal association does not have any implications for prevention of diabetes. This needs to be mentioned in the Discussion section.

Reply: We agree with the comments that remnant cholesterol/triglycerides might not be mechanistically involved in the pathogenesis of diabetes. As suggested, we have added the interpretation for the association between remnant cholesterol/triglycerides and type 2 diabetes in the Discussion section as follows: "Notwithstanding we showed a causal link between triglyceride-rich lipoproteins and type 2 diabetes risk in this study, elevated remnant cholesterol and triglycerides were unlikely to be mechanistically involved in the pathogenesis of type 2 diabetes. Seemingly, the genetic associations between remnant cholesterol, triglycerides, and type 2 diabetes could be attributed to increased likelihood of diabetes detection and diagnosis in participants with elevated triglycerides as clinicians often use elevated triglycerides as a biomarker of insulin resistance. We also showed that baseline serum remnant cholesterol and triglycerides were associated with higher risks of progression from ischemic heart disease to IHD-T2D multimorbidity but not the progression from type 2 diabetes to IHD-T2D multimorbidity, suggesting differential roles of triglyceride-rich lipoproteins in disease progression among patients that developed heart disease or diabetes."

Indeed, pooling of ischemic heart disease, stroke, and diabetes is difficult to make sense of in genetic causal analyses. Therefore, please also present sensitivity analyses only pooling ischemic heart disease and stroke, which from a causal perspective would make much more sense as elevated remnant cholesterol/triglycerides are causal risk factors for both these diseases. This analyses

preferably should also be presented in the main part of the paper but could also be presented in the supplementary section.

Reply: We agree with the comments that only pooling the three diseases together might be difficult to make sense of genetic causal analysis. As suggested, we have re-analysed the data treating the three cardiometabolic disease as separate endpoints but found no significant association between remnant cholesterol/triglycerides and stroke (please see Supplementary Table 3 and Supplementary Fig. 3). Alternatively, we developed a second multistate model allowing the transition from baseline to ischemic heart disease/type 2 diabetes and to subsequent IHD-T2D multimorbidity as follows (Supplementary Fig. 1b):

Briefly, we showed that serum remnant cholesterol/triglycerides were associated with higher risks of progression from ischemic heart disease to IHD-T2D multimorbidity. Mendelian randomization analysis also demonstrated a causal link between remnant cholesterol/triglycerides and IHD-T2D multimorbidity. We have presented these results in Results section (Table 3, Figure 1, and Figure 2) and Supplementary materials (Supplementary Table 4).

Introduction: “Cardiometabolic multimorbidity” should be defined. The aim of the article does not seem to be to “compare” observational and genetic analyses, but rather to test if elevated remnant cholesterol and triglycerides are associated with increased risk of cardiometabolic multimorbidity (using observational and genetic analyses). The aim should be rephrased to harmonize with the results and conclusion.

Reply: As suggested, we added a clearer definition of cardiometabolic multimorbidity in the Introduction section as follows: “cardiometabolic multimorbidity, characterized by the concurrence of at least two cardiometabolic diseases”. In addition, we have revised the aim to harmonize with the results and conclusion as follows: “In this study, we utilized multistate modelling, two-stage least squares regression-based Mendelian randomization, doubly-ranked stratification-based non-linear Mendelian randomization to investigate the associations between remnant cholesterol, triglycerides and the risks of cardiometabolic multimorbidity in the UK Biobank.”

It is not completely clear if users of statins were included or excluded from the various analyses performed (please mention that where relevant in the paper). As high risk of cardiovascular disease and diabetes status influences the clinical decision to prescribe statins, statin use is a likely collider on the pathway from remnant cholesterol to cardiometabolic risk. Therefore, (if you have not already done it) please perform sensitivity analyses where individuals on statins are excluded in observational

analyses and in the first stage regression in genetic analyses, but not excluded from the second-stage regression in Mendelian randomization. These sensitivity analyses can be presented in the Supplementary, with the findings here from summarized in the Results section of the main paper. Please also mention in the limitations section of the Discussion, that statin use is a likely collider on the pathway from remnant cholesterol to cardiometabolic risk, and what that means for the present results.

Reply: In this study, we excluded participants receiving lipid-lowering medications (n = 98,828) in both observational analysis and one-sample Mendelian randomization analysis. We have added a new participant flow chart in the revised manuscript (Supplementary Fig. 2). As suggested, we also performed sensitivity analysis including participants receiving lipid-lowering medications in the second-stage regression in Mendelian randomization. Results are attached below:

We have mentioned this in the limitation in the Discussion section as follows: “Sixth, cardiometabolic health status could influence clinical decision to prescribe lipid-lowering medication, which might a collider on the pathway from remnant cholesterol and triglycerides to cardiometabolic risk, leading to biased risk estimates. However, sensitivity analysis including participants receiving lipid-lowering medication at the second-stage regression yielded similar results compared to the main analysis.”

Results: Separate analyses of the endpoints included in the composite endpoint (ischemic heart disease, stroke, and diabetes) should be done, otherwise it cannot be distinguished if the results are all driven by an association with ischemic heart disease (which is already well known), an association with stroke, and/or an association with diabetes (which as mentioned above in contrast to ischemic heart disease and stroke, likely is not mechanistic, but diagnostic).

Reply: As suggested, we have analysed the associations between remnant cholesterol, triglycerides,

and separate cardiometabolic disease endpoints. We found significant associations between remnant cholesterol/triglycerides and ischemic heart disease and type 2 diabetes. Please see the figure below and Supplementary Table 3.

Results: Non-linear Mendelian randomization is a method which currently (if it is used) may not be advisable, and therefore should be used with extreme caution. As exemplified with the article on Vitamin D levels (PMID: 34717822), there are risks of unexpected biases (PMID: 36528346); tools for addressing these have not yet been developed. In the present paper, non-linear Mendelian randomization seems somewhat redundant and could be dropped; otherwise, extensive sensitivity analyses are required, including using the doubly-ranked method (PMID: 36528346) and estimation of associations between the genetic risk score and measured confounders in each of the strata of exposure residuals.

Reply: We agree that residual-based stratification might lead to biased effect estimates. As suggested, we used the new doubly-ranked stratification-based method, which could relax the strong parametric assumptions of linearity and homogeneity between the instrument and the exposure to form the strata in residual-based stratification. We found gradually increased risks of cardiometabolic morbidity and IHD-T2D multimorbidity in relation to higher remnant cholesterol and triglycerides from the linear Mendelian randomization analysis, while the linear trends of the causal associations slightly attenuated at higher concentrations for both remnant cholesterol and triglyceride. Please see the figure (Figure 2 in the manuscript) below.

Minor comments:

Methods: Were the same individuals included in both observational and genetic analyses?

Reply: We employed different participant inclusion criteria for observational and Mendelian randomization analysis as follows: “In the observational analysis, participants were excluded if they had any cardiometabolic diseases (type 2 diabetes, ischemic heart disease, and stroke) at baseline, were prescribed lipid-lowering medications (n = 98,828, see Supplementary Methods for details), or had missing values for remnant cholesterol (n = 73,390) or triglycerides (n = 33,284). Therefore, 334,030 and 365,577 eligible participants were included in the observational analysis of remnant cholesterol and triglycerides, respectively (Supplementary Fig. 2a). In the Mendelian randomization analysis, participants were excluded if they were not white British (n = 59,895), did not pass the quality control for genotyping (n = 3636), were prescribed lipid-lowering medications (n = 98,828), or had missing values for remnant cholesterol (n = 73,390) or triglycerides (n = 33,284). Therefore, 301,565 and 330,031 eligible participants were included in the Mendelian randomization analysis of remnant cholesterol and triglycerides, respectively (Supplementary Fig. 2b).” We also added a new participant flow chart (Supplementary Fig. 2):

Abstract: “Cardiometabolic disease event”, and “multimorbidity” should be defined in Methods. The event of interest in the Mendelian randomization analyses should also be specified.

Reply: We have added clearer definitions of “cardiometabolic disease” and “cardiometabolic multimorbidity” in the Methods section as follows: “The prevalent and newly on-set cardiometabolic diseases, namely type 2 diabetes, ischemic heart disease, and stroke, were identified via linkage to primary care records, hospital inpatient records, and death registry records and outcomes were coded using the International Classification of Disease, 10th revision codes: type 2 diabetes, E11; ischemic heart disease, I21-I25; stroke, I60-I64. We censored participants at the end of follow-up, at the date of the first occurrence of any cardiometabolic disease or development of multimorbidity, and at the date of loss to follow-up, whichever was first. Cardiometabolic multimorbidity was defined as the occurrence of at least two cardiometabolic diseases, like the co-existence of ischemic heart disease and type 2 diabetes (IHD-T2D multimorbidity) and the co-existence of stroke and ischemic heart disease (IHD-stroke multimorbidity).”

Reviewer #2 (Remarks to the Author):

The manuscript by Zhao Y., et al. studies the associations between elevated remnant cholesterol and triglycerides levels with the risk of developing cardiometabolic multimorbidity from the UK Biobank encompassing over 370,000 individuals. The results suggest that high levels of these two markers are associated with an increased risk of developing multiple cardiometabolic diseases. In particular, triglyceride-rich lipoproteins could be both potential risk factors and therapeutic targets for the prevention and treatment of cardiometabolic diseases.

The manuscript is overall well-written and organized, however, several points should be better discussed and clarified.

- Did the study account for any lifestyle or environmental factors that could influence the progression of cardiometabolic diseases to multimorbidity?

Reply: As suggested, we derived a diet quality score and a sleep quality score as indicators for healthy diet and healthy sleep pattern, respectively. Please see the Supplementary methods: “Diet was assessed via a food frequency questionnaire at baseline (<https://biobank.ndph.ox.ac.uk/showcase/label.cgi?id=100052>). We calculated the diet quality score based on the intake of fruits, vegetables, processed meat, red meat, fish, whole grains, and refined grains. One point was assigned to each food item if: the average daily intake of: 1) fruits ≥ 3 servings/day; 2) vegetables ≥ 3 servings/day; 3) fish ≥ 2 servings/week; 4) processed meat ≤ 1 servings/week; 5) red meat ≤ 1.5 servings/week; 6) whole grain ≥ 3 servings/week; 7) refined grain ≤ 1.5 servings/week. The diet quality score ranged from 0 to 7 with a higher score indicating a better diet quality. A higher diet quality score indicated better adherence to a healthy dietary pattern.” and “Sleep pattern was evaluated via a touchscreen questionnaire during the baseline assessment (<https://biobank.ndph.ox.ac.uk/showcase/label.cgi?id=100057>). We derived the sleep quality score based on chronotype, sleep duration, frequency of insomnia, snoring, and daytime sleepiness. One point was assigned to each item if: 1) early chronotype; 2) sleep 7 to 8 hours per day; 3) never or rarely had insomnia symptoms; 4) no self-reported snoring; 5) no frequent daytime sleepiness.

Therefore, the sleep quality score ranged from 0 to 5 with a higher score indicating healthier sleep pattern.” Subgroup analysis suggested that the associations between remnant cholesterol, triglycerides, and cardiometabolic multimorbidity were generally similar when stratified by lifestyle factors including diet, sleep, alcohol intake, smoking status, and physical activity. Please see Supplementary Table 4.

- How was the progression from a first cardiometabolic disease event to multimorbidity defined? Were there any specific criteria or time intervals used to identify multimorbidity?

Reply: We have added more clearer definitions of cardiometabolic disease and cardiometabolic multimorbidity in the Methods section as follows: “The prevalent and newly on-set cardiometabolic diseases, namely type 2 diabetes, ischemic heart disease, and stroke, were identified via linkage to primary care records, hospital inpatient records, and death registry records and outcomes were coded using the International Classification of Disease, 10th revision codes: type 2 diabetes, E11; ischemic heart disease, I21-I25; stroke, I60-I64. We censored participants at the end of follow-up, at the date of the first occurrence of any cardiometabolic disease or development of multimorbidity, and at the date of loss to follow-up, whichever was first. Cardiometabolic multimorbidity was defined as the concurrence of at least two cardiometabolic diseases, like the co-existence of ischemic heart disease and type 2 diabetes (IHD-T2D multimorbidity) and the co-existence of stroke and ischemic heart disease (IHD-stroke multimorbidity).” There were no specific time intervals for identifying multimorbidity.

- The abstract reports "a median follow-up of 12.6 years", but what is the span? Can the results be affected if adjusting for the years passed between baseline and follow-up?

Reply: The median follow-up time was derived treating any first cardiometabolic disease as the endpoint. Because the times intervals differ between each transition, we did not include the time interval as a covariate in the multistate model. Instead, we used age as the time-varying covariate in the model in this study.

- While the introduction briefly mentions previous studies and their findings regarding the association of triglycerides with cardiovascular disease, it would benefit from a more comprehensive review of the existing literature, including any conflicting or inconclusive results. For instance:

<https://doi.org/10.1016/j.atherosclerosis.2022.06.205>, <https://doi.org/10.3389/fgene.2023.1117778>,
<https://doi.org/10.1016/j.numecd.2011.06.003>, <https://doi.org/10.3390/nu13061935>

Reply: As suggested, we have added recent studies concerning the association of triglycerides with cardiovascular disease and cited the recommended references in the Introduction section as follows: “It is noteworthy that diet, like fat and cholesterol intake, and lifestyle factors might substantially influence triglyceride-rich lipoprotein metabolism.¹⁹⁻²¹ The complex interplay between lipid species might also confound the observational relationships between triglyceride-rich lipoproteins and cardiometabolic diseases.²²”

- What are the potential implications for clinical practice, and what further research is needed to evaluate the effectiveness and feasibility of interventions aimed at reducing triglyceride-rich lipoproteins? This should probably be better discussed.

Reply: As suggested, we have added more discussion concerning the clinical implications of our study in the Discussion section as follows: “Notwithstanding we showed a causal link between

triglyceride-rich lipoproteins and type 2 diabetes risk in this study, elevated remnant cholesterol and triglycerides were unlikely to be mechanistically involved in the pathogenesis of type 2 diabetes. Seemingly, the genetic associations between remnant cholesterol, triglycerides, and type 2 diabetes could be attributed to increased likelihood of diabetes detection and diagnosis in participants with elevated triglycerides as clinicians often use elevated triglycerides as a biomarker of insulin resistance.³³ We also showed that baseline serum remnant cholesterol and triglycerides were associated with higher risks of progression from ischemic heart disease to IHD-T2D multimorbidity but not the progression from type 2 diabetes to IHD-T2D multimorbidity, suggesting differential roles of triglyceride-rich lipoproteins in disease progression among patients that developed heart disease or diabetes.” and “In conclusion, remnant cholesterol and triglycerides were both observationally and genetically linked to higher cardiometabolic multimorbidity risks. These findings imply that triglyceride-rich lipoproteins could serve potential risk factors and therapeutic targets for the prevention and treatment of cardiometabolic diseases, which necessitate further validation through randomized controlled trials focusing on triglyceride-rich lipoproteins.”.

- Since alcohol was estimated from food frequency questionnaires, why were no other dietary factors considered in the analysis?

Reply: As suggested, we derived a diet quality score as an indicator of healthy diet quality as follows: “Diet was assessed via a food frequency questionnaire at baseline (<https://biobank.ndph.ox.ac.uk/showcase/label.cgi?id=100052>). We calculated the diet quality score based on the intake of fruits, vegetables, processed meat, red meat, fish, whole grains, and refined grains. One point was assigned to each food item if: the average daily intake of: 1) fruits ≥ 3 servings/day; 2) vegetables ≥ 3 servings/day; 3) fish ≥ 2 servings/week; 4) processed meat ≤ 1 servings/week; 5) red meat ≤ 1.5 servings/week; 6) whole grain ≥ 3 servings/week; 7) refined grain ≤ 1.5 servings/week. The diet quality score ranged from 0 to 7 with a higher score indicating a better diet quality. A higher diet quality score indicated better adherence to a healthy dietary pattern.” We have included the diet quality score as a covariate in the multistate model.

- The study adjusted for many covariates, but it did not incorporate dietary information than alcohol. Triglyceride levels can be heavily influenced by diet and lifestyle factors, such as the consumption of high-fat and high-sugar foods. What could be the impact of dietary information on the results?

Reply: As suggested, we derived a diet quality score as an indicator of healthy diet quality as follows: “Diet was assessed via a food frequency questionnaire at baseline (<https://biobank.ndph.ox.ac.uk/showcase/label.cgi?id=100052>). We calculated the diet quality score based on the intake of fruits, vegetables, processed meat, red meat, fish, whole grains, and refined grains. One point was assigned to each food item if: the average daily intake of: 1) fruits ≥ 3 servings/day; 2) vegetables ≥ 3 servings/day; 3) fish ≥ 2 servings/week; 4) processed meat ≤ 1 servings/week; 5) red meat ≤ 1.5 servings/week; 6) whole grain ≥ 3 servings/week; 7) refined grain ≤ 1.5 servings/week. The diet quality score ranged from 0 to 7 with a higher score indicating a better diet quality. A higher diet quality score indicated better adherence to a healthy dietary pattern.” Indeed, we found that participants with higher serum remnant cholesterol/triglycerides tend to have a less healthy diet quality (Table 1). We have included the diet quality score as a covariate in the multistate model. Subgroup analysis stratified by diet quality score suggested the associations between remnant cholesterol/triglycerides and IHD-T2D multimorbidity did not differ significantly among participants with

a diet quality score ≥ 4 or < 3 (Supplementary Table 4).

- The authors in the "Discussion" mention: "Besides, remnant cholesterol was associated with risks of developing diabetes after transplantation in renal transplant recipients. Our results were consistent with these findings", but this is not clear how the presented results are supporting this statement, please clarify.

Reply: In the revised manuscript, we have removed this sentence in order to focus the discussion more on the associations between remnant cholesterol/triglycerides and cardiometabolic multimorbidity.

- How were the instrumental variables for the Mendelian randomization analysis selected? Were they associated with the exposure only or also independent of potential confounders?

Reply: The instrumental variables for Mendelian randomization analysis were selected as follows (please see the Methods section): "To avoid pleiotropy of genetic instruments, we retrieved the 16 gain-of-function and loss-of-function single nucleotide polymorphisms in a previous study by Kalltoft et al.⁵ Candidate genes of these SNPs encode key enzymes, regulatory factors, and lipoproteins that have well-established effects on triglyceride-rich lipoprotein metabolism, including LPL, angiopoietin-like 3 (ANGPTL3), angiopoietin-like 4 (ANGPTL4), apolipoprotein C-III (APOC3), and apolipoprotein A-V (APOAV)^{7,40,41}". Such genetic instrument selection strategy was comparable to that used in drug target Mendelian randomization analysis, which could reduce the possibility of horizontal pleiotropy and weak instrument bias. Thus, the instrumental variables were unlikely to be associated with confounding factors.

Reviewer #3 (Remarks to the Author):

With interest I've read the paper Elevated blood remnant cholesterol and triglycerides are causally related to the risks of cardiometabolic multimorbidity, by Zhao et al. The authors aimed to assess the observational and causal link between remnant cholesterol and triglycerides with the risks of cardiometabolic multimorbidity. Cardiometabolic morbidity was defined as the concomitant prevalence of two+ cardiometabolic diseases such as T2D, IHD and stroke. The authors performed analyses in a subset of the UK Biobank. The authors only investigated incident cases, and excluded participants with prior cardiometabolic disease. In 371,234 participants, they quantified a total of 39,805 incident cases of cardiometabolic disease, of which 15,547 progressed to cardiometabolic multimorbidity. Using these data and by employing various models, the authors showed that remnant cholesterol and triglycerides both were associated with increased risk of incident cardiometabolic disease and with progression to multimorbidity. They also showed that these associations seem causal, as quantified by both linear and nonlinear Mendelian Randomization.

Although these results are intuitive, I commend the authors for quantifying these associations. There are a few comments and questions, especially with regards to MR:

Methods:

1. GRS: internally weighted, this does increase the risk for weak instrument bias (also see 2). For TG at least, there is the option of validating results using externally weighed GWAS values. Suggest

elaborating on this limitation and performing sensitivity analyses with externally weighed SNPs. (e.g. using the weights from Kalsoft et al).

Reply: As suggested, the GRS was re-calculated using externally weighed 13 SNPs and related weights (please see Supplementary Table 7) from Kalsoft et al (PMID: 32267934) in the revised manuscript.

2. Internally weighted, some instruments are invalid (as P-value for some of the instruments > 5e-4). Suggest performing MR-egger and perform analyses where weak SNPs are excluded.

Reply: As mentioned above, we re-calculated the GRS using externally weighed 13 SNPs and related weights (please see Supplementary Table 7) from Kalsoft et al (PMID: 32267934) in the revised manuscript. All the 13 SNPs are gain-of-function and loss-of-function SNPs, the candidate genes of which encode key enzymes, regulatory factors, and lipoproteins that have well-established effects on triglyceride-rich lipoprotein metabolism. We have introduced instrumental variable selection as follows (please see the Methods section): “To avoid pleiotropy of genetic instruments, we retrieved the 16 gain-of-function and loss-of-function single nucleotide polymorphisms in a previous study by Kalsoft et al.⁵ Candidate genes of these SNPs encode key enzymes, regulatory factors, and lipoproteins that have well-established effects on triglyceride-rich lipoprotein metabolism, including LPL, angiotensin-like 3 (ANGPTL3), angiotensin-like 4 (ANGPTL4), apolipoprotein C-III (APOC3), and apolipoprotein A-V (APOAV)^{7,40,41}”.

3. Current MR studies are advised to follow the STROBE-MR guidelines and include the checklist in their supplementals.

Reply: As suggested, we have added a STROBE-MR checklist at the end of the Supplementary material.

5. The use of ICD-10 codes ensures that the outcome is relatively clean from bias. However, The inclusion of I20 might be a bit problematic, as this refers to Angina pectoris, including many non-specified. This increases the risk of including patients with AP symptoms that refer to something different than purely cardiac. Please consider performing sensitivity analyses with only 'hard' coronary artery disease (e.g. excluding I20).

Reply: As suggested, ischemic heart disease was identified using ICD-10 I21-I25, excluding I20, in the revised manuscript.

6. Potentially interesting: Progression from which to what? E.g. DM2 to ischemic heart disease? Suggest splitting it out.

Reply: In the revised manuscript, we added more clearer definitions of the disease transition model as follows: “In this study, we defined three states of participants, namely free of cardiometabolic disease, incidence of first cardiometabolic disease, and development of cardiometabolic morbidity, allowing two disease transitions, namely 1) from baseline to first cardiometabolic disease and 2) from first cardiometabolic disease to multimorbidity (Supplementary Fig. 1a).” and “In addition, we also constructed a multistate model that also included ischemic heart disease and type 2 diabetes allowing four transitions, namely 1) from baseline to incident type 2 diabetes, 2) from baseline to ischemic heart disease, 3) from type 2 diabetes to IHD-T2D multimorbidity, and 4) from ischemic heart disease to IHD-T2D multimorbidity.” We also added a new figure illustrating the disease transition model (Supplementary Fig. 1):

7. The authors excluded participants with cardiovascular disease before inclusion date, which could introduce participation bias in their MR study. (MR is lifetime risk on CVD, so excluding participants based on disease before inclusion in the biobank will likely result in skewed results). How does this affect your results and how large is this issue? Would reanalysis with participants with CVD before inclusion alter the results?

Reply: As suggested, we included participants with existing cardiometabolic disease at baseline in the Mendelian randomization analysis in the revised manuscript as follows (please see the Methods section): “In the Mendelian randomization analysis, participants were excluded if they were not white British (n = 59,895), did not pass the quality control for genotyping (n = 3636), were prescribed lipid-lowering medications (n = 98,828), or had missing values for remnant cholesterol (n = 73,390) or triglycerides (n = 33,284). Therefore, 301,565 and 330,031 eligible participants were included in the Mendelian randomization analysis of remnant cholesterol and triglycerides, respectively (Supplementary Fig. 2b).” Besides, we also added a new participant flow chart:

The results remained similar compared to the previous Mendelian randomization analysis which excluded participants with cardiometabolic disease at baseline.

Results:

1. Harrell's C index did significantly increase, but does numerically not add any value. Suggest rewriting as the added clinical value is extremely limited (or even absent).

Reply: We acknowledge that the slight but statistically significant increases in Harrell's C index might be difficult to interpret in the context of a multistate model. We have mentioned this in the limitation section of the Discussion in the revised manuscript as follows: "Fifth, although the Harrell's C index significantly increased after adding remnant cholesterol and triglycerides into the basal model, a risk prediction model specific for developing cardiometabolic morbidity is still lacking. Also, it remains to be studied whether remnant cholesterol and triglycerides could improve the discrimination and calibration power of the risk prediction model for progression from first cardiometabolic disease to multimorbidity."

Discussion:

1. The discussion is rather underwhelming. Suggest expanding upon the novelties of this study, and expand on non-linear relationships for example.

Reply: As suggested, we added more discussion concerning the clinical implications and study limitations as follows: "Notwithstanding we showed a causal link between triglyceride-rich lipoproteins and type 2 diabetes risk in this study, elevated remnant cholesterol and triglycerides were unlikely to be mechanistically involved in the pathogenesis of type 2 diabetes. Seemingly, the genetic associations between remnant cholesterol, triglycerides, and type 2 diabetes could be attributed to increased likelihood of diabetes detection and diagnosis in participants with elevated triglycerides as clinicians often use elevated triglycerides as a biomarker of insulin resistance.³² We also showed that baseline serum remnant cholesterol and triglycerides were associated with higher risks of progression from ischemic heart disease to IHD-T2D multimorbidity but not the progression from type 2 diabetes to IHD-T2D multimorbidity, suggesting differential roles of triglyceride-rich lipoproteins in disease progression among patients that developed heart disease or diabetes." and "Fifth, although the Harrell's C index significantly increased after adding remnant cholesterol and triglycerides into the basal model, a risk prediction model specific for developing cardiometabolic morbidity is still lacking. Also, it remains to be studied whether remnant cholesterol and triglycerides could improve the discrimination and calibration power of the risk prediction model for progression from first cardiometabolic disease to multimorbidity. Sixth, cardiometabolic health status could influence clinical decision to prescribe lipid-lowering medication, which might a collider on the pathway from remnant cholesterol and triglycerides to cardiometabolic risk, leading to biased risk estimates.^{35,36} However, sensitivity analysis including participants receiving lipid-lowering medication at the second-stage regression yielded similar results compared to the main analysis."

Minor comments:

- Some minor language errors (line 78 for example, 349-352)

Reply: As suggested, we have thoroughly revised the manuscript and corrected several grammatical errors.

REVIEWER COMMENTS

Reviewer #1 (Remarks to the Author):

Zhao et al. have diligently revised the manuscript in response to my comments. In my view, this interesting manuscript has greatly improved. I have a few additional comments.

Major comments:

1. Abstract: The end of the sentence "...particularly the progression of ischemic heart disease to the multimorbidity of ischemic heart disease and type 2 diabetes" coupled with the next sentence. "... These results advocate for effective management of remnant cholesterol and triglycerides as a potential strategy in mitigating the risks of progression from single cardiometabolic disease to multimorbidity" makes it sound like remnant cholesterol and triglycerides can be managed to mitigate risk of progression to type 2 diabetes. As the authors adequately addressed in the Discussion, lowering of remnant cholesterol and triglycerides may simply lower the chance of having an existing type 2 diabetes diagnosed; to me, this does seem like a fruitful strategy for preventing multimorbidity. Please revise.
2. Discussion: The same point goes for the conclusion; when viewed together with previous evidence, the results of the current study do probably not imply that triglyceride-rich lipoproteins may be beneficial therapeutic targets for prevention and treatment of cardiometabolic disease (IHD, stroke + diabetes), but probably only for cardiovascular disease (IHD and ischemic stroke). Please revise.
3. As mentioned, statin use is a collider on the pathway from remnant cholesterol to risk of atherosclerotic cardiovascular diseases. There is no way to completely solve this problem; however, excluding individuals with statin use from the second stage of the Mendelian randomization regression may exacerbate bias rather than ameliorate it. I therefore kindly suggest that the main analyses retains users of lipid-lowering therapy, either only in the first stage of the regression or in both stages, or alternatively, other methods to deal with this issue may be found in a recent publication (PMID: 36821633) from experts in Mendelian randomization which states "... Conditioning on statin use via exclusions/stratifications or adjustment would be problematic, however, as this could heighten the potential for index event bias given the likely role of statin use as a mediator".
4. I insist that non-linear Mendelian randomization, even though an important area of methodological research, is not mature yet to be applied. See the current publication PMID:37845826, in which the Authors found a causal non-linear effect of body mass index on sex (!) using both doubly ranked and residual based non-linear Mendelian randomization. Of course, sex was used as a negative control and such a causal effect is completely nonsensical, pointing to yet unresolved high risk of bias with non-linear

Mendelian randomization. I reiterate my view that this Manuscript does not require non-linear Mendelian randomization to be of interest, so the simplest solution is for this analysis to be dropped. If the Authors are very keen to keep this analysis despite the severe risk of bias, estimation of the associations between the genetic risk score and negative controls should be done in each non-linear stratum, to show that there are no associations with negative controls such as sex, age, and socioeconomic status (i.e. signs of population stratification) that could lead to differential results by strata. If there are differential associations between the genetic risk score with negative controls in the strata, it is a sign that the non-linear MR analysis is biased.

Minor comments:

5. Logistic regression, which was used for Mendelian randomization, is a log-scale regression. Therefore, a “linear” logistic regression will have a logarithmic shape when plotted with an odds ratio y-axis without transformation. In the current form, the Figure 2 may therefore very well depict linear associations to me. To enable visual evaluation of non-linearity, the y-axis of Figure 2 should be log-transformed with the natural logarithm (the labels from 0.8-1.6 in steps of 0.2 can still be retained, but they will not be equally distanced from each other. However, as mentioned in the previous comment, Figure 2 could simply be dropped.

Reviewer #2 (Remarks to the Author):

The authors addressed all raised points in their revision.

Reviewer #3 (Remarks to the Author):

My questions have been answered sufficiently.

I would like to note that, from what I understand from the exclusion criteria described by the authors, excluding lipid-lowering-therapy-using participants from the MR analyses in this case introduced collider bias. Reviewer 1 did describe that they should not exclude these participants for the second stage of their regression. It is not entirely clear to me if they have done so.

REVIEWER COMMENTS

Reviewer #1 (Remarks to the Author):

Zhao et al. have diligently revised the manuscript in response to my comments. In my view, this interesting manuscript has greatly improved. I have a few additional comments.

Major comments:

1. Abstract: The end of the sentence "...particularly the progression of ischemic heart disease to the multimorbidity of ischemic heart disease and type 2 diabetes" coupled with the next sentence "... These results advocate for effective management of remnant cholesterol and triglycerides as a potential strategy in mitigating the risks of progression from single cardiometabolic disease to multimorbidity" makes it sound like remnant cholesterol and triglycerides can be managed to mitigate risk of progression to type 2 diabetes. As the authors adequately addressed in the Discussion, lowering of remnant cholesterol and triglycerides may simply lower the chance of having an existing type 2 diabetes diagnosed; to me, this does seem like a fruitful strategy for preventing multimorbidity. Please revise.

Reply: As suggested, we have revised the last sentences of Abstract as follows: "These results advocate for effective management of remnant cholesterol and triglycerides as a potential strategy in mitigating the risks of cardiometabolic multimorbidity."

2. Discussion: The same point goes for the conclusion; when viewed together with previous evidence, the results of the current study do probably not imply that triglyceride-rich lipoproteins may be beneficial therapeutic targets for prevention and treatment of cardiometabolic disease (IHD, stroke + diabetes), but probably only for cardiovascular disease (IHD and ischemic stroke). Please revise.

Reply: As suggested, we have revised the first sentence of Discussion as follows: "In this study of over 300,000 UK Biobank participants, serum remnant cholesterol and triglycerides were significantly associated with higher risks of cardiometabolic multimorbidity, particularly the risks of the progression from ischemic heart disease to IHD-T2D multimorbidity." In addition, we also revised the last two sentences of the second paragraph of Discussion as follows: "We further demonstrated a markedly higher risk for progression from initial ischemic heart disease to IHD-T2D multimorbidity using a multistate model. In this regard, our results provide evidence endorsing the treatment of blood triglyceride-rich lipoproteins in people with existing cardiovascular diseases to mitigate the risks of multimorbidity progression."

3. As mentioned, statin use is a collider on the pathway from remnant cholesterol to risk of atherosclerotic cardiovascular diseases. There is no way to completely solve this problem; however, excluding individuals with statin use from the second stage of the Mendelian randomization regression may exacerbate bias rather than ameliorate it. I therefore kindly suggest that the main analyses retains users of lipid-lowering therapy, either only in the first stage of the regression or in both stages, or alternatively, other methods to deal with this issue may be found in a recent publication (PMID: 36821633) from experts in Mendelian randomization which states "... Conditioning on statin use via exclusions/stratifications or adjustment would be

problematic, however, as this could heighten the potential for index event bias given the likely role of statin use as a mediator”.

Reply: As suggested, we have included participants that used lipid-lowering medications in both first- and second-stage of Mendelian randomization in the main analysis of the revised manuscript. The results did not change significantly. Please see Figure 1 and Supplementary Fig. 3 (attached below).

Figure 1

Supplementary Fig. 3

4. I insist that non-linear Mendelian randomization, even though an important area of methodological research, is not mature yet to be applied. See the current publication PMID:37845826, in which the Authors found a causal non-linear effect of body mass index on sex (!) using both doubly ranked and residual based non-linear Mendelian randomization. Of course, sex was used as a negative control and such a causal effect is completely nonsensical, pointing to yet unresolved high risk of bias with non-linear Mendelian randomization. I reiterate my view that this Manuscript does not require non-linear Mendelian randomization to be of interest, so the simplest solution is for this analysis to be dropped. If the Authors are very keen to keep this analysis despite the severe risk of bias, estimation of the associations between the genetic risk

score and negative controls should be done in each non-linear stratum, to show that there is no associations with negative controls such as sex, age, and socioeconomic status (i.e. signs of population stratification) that could lead to differential results by strata. If there are differential associations between the genetic risk score with negative controls in the strata, it is a sign that the non-linear MR analysis is biased.

Reply: As suggested, we have dropped all non-linear MR analysis in the revised manuscript to avoid potential concerns regarding methodological validity.

Minor comments:

5. Logistic regression, which was used for Mendelian randomization, is a log-scale regression. Therefore, a “linear” logistic regression will have a logarithmic shape when plotted with an odds ratio y-axis without transformation. In the current form, the Figure 2 may therefore very well depict linear associations to me. To enable visual evaluation of non-linearity, the y-axis of Figure 2 should be log-transformed with the natural logarithm (the labels from 0.8-1.6 in steps of 0.2 can still be retained, but they will not be equally distanced from each other. However, as mentioned in the previous comment, Figure 2 could simply be dropped.

Reply: As suggested, we have dropped all non-linear MR analysis, including Figure 2, in the revised manuscript to avoid potential concerns regarding methodological validity.

Reviewer #2 (Remarks to the Author):

The authors addressed all raised points in their revision.

Reviewer #3 (Remarks to the Author):

My questions have been answered sufficiently.

I would like to note that, from what I understand from the exclusion criteria described by the authors, excluding lipid-lowering-therapy-using participants from the MR analyses in this case introduced collider bias. Reviewer 1 did describe that they should not exclude these participants for the second stage of their regression. It is not entirely clear to me if they have done so.

Reply: As suggested, we have included participants that used lipid-lowering medications in both first- and second-stage of Mendelian randomization in the main analysis of the revised manuscript. The results did not change significantly.

REVIEWERS' COMMENTS

Reviewer #1 (Remarks to the Author):

The authors have adressed all my comments, I thank the authors for their work to adress my concerns.